# Improving the management and security of COVID 19 diagnostic test data with a digital platform in resource-limited settings: The case of PlaCARD in Cameroon

**Jules Brice Tchatchueng-Mbougua**[1,2]*, **Loique Landry Messanga Essengue**[1], **Francis Jaudel Septoh Yuya**[1], **Vanessa Kamtchogom**[1], **Achta Hamadou**[1,3], **Serge Alain Sadeuh-Mbah**[4], **Paul Alain Tagnouokam-Ngoupo**[4], **Maurice Tchuente**[5], **Richard Njouom**[4], **Sara Eyangoh**[3,6], **Mathurin Cyrille Tejiokem**[1]

**1** Epidemiology and public Heath Unit, Centre Pasteur du Cameroun, membre du Réseau International des Instituts Pasteur (RIIP), Yaounde, Cameroon, **2** IRD UMI 209 UMMISCO, University of Yaounde I, P.O. Box 337 Yaounde, Cameroon, **3** Public Health Emergency Operations Coordination Center, Yaounde, Cameroon, **4** Virology Unit, Centre Pasteur du Cameroun, membre du Réseau International des Instituts Pasteur (RIIP), Yaounde, Cameroon, **5** Fondation pour la recherche l'ingénierie et l'innovation, Yaounde, Cameroon, **6** Mycobacteriology Unit, Centre Pasteur du Cameroun, membre du Réseau International des Instituts Pasteur (RIIP), Yaounde, Cameroon

* tchatchueng@pasteur-yaounde.org

## Abstract

During the COVID 19 pandemic, round-the-clock demand for COVID -19 laboratory tests exceeded capacity, placing a significant burden on laboratory staff and infrastructure. The use of laboratory information management systems (LIMS) to streamline all phases of laboratory testing (preanalytical, analytical, and postanalytical) has become inevitable. The objective of this study is to describe the architecture, implementation, and requirements of PlaCARD, a software platform for managing patient registration, medical specimens, and diagnostic data flow, as well as reporting and authentication of diagnostic results during the 2019 coronavirus pandemic (COVID -19) in Cameroon. Building on its experience with biosurveillance, CPC developed an open-source, real-time digital health platform with web and mobile applications called PlaCARD to improve the efficiency and timing of disease-related interventions. PlaCARD was quickly adapted to the decentralization strategy of the COVID 19 testing in Cameroon and, after specific user training, was deployed in all COVID 19 diagnostic laboratories and the regional emergency operations center. Overall, 71% of samples tested for COVID 19 by molecular diagnostics in Cameroon from 05 March 2020 to 31 October 2021 were entered into PlaCARD. The median turnaround time for providing results was 2 days [0–2.3] before April 2021 and decreased to 1 day [1– 1] after the introduction of SMS result notification in PlaCARD. The integration of LIMS and workflow management into a single comprehensive software platform (PlaCARD) has strengthened COVID 19 surveillance capabilities in Cameroon. PlaCARD has demonstrated that it can be used as a LIMS for managing and securing test data during an outbreak.

**Data Availability Statement:** The goal of this article was not to analyse the data generated by the platform, but to demonstrate the feasibility of using the PlaCARD platform in an epidemic situation. Therefore, the data generated for the platform were minimally illustrated in the paper.

**Funding:** The authors received no specific funding for this work.

**Competing interests:** The authors have declared that no competing interests exist.

## Author summary

Following the emergence of a novel coronavirus (SARS-CoV-2) in Wuhan, China, in December 2019 and its global spread, the number of tests conducted by laboratories COVID 19 increased rapidly, placing a significant burden on laboratory staff and infrastructure. This resulted in long turnaround times for laboratory results, which in turn led to delays in immediate follow-up and strategic decision making. The use of laboratory information management systems (LIMS) to streamline all phases of laboratory testing (preanalytical, analytical, and postanalytical) has therefore become essential. The objective of this study was to describe thearchitecture requirements of the PlaCARD (Platform for Collecting, Analyzing, and Reporting Data) digital platform and its implementation for decentralization of COVID 19 diagnostics in a resource-constrained environment. The results of this study showed that integrating of the LIMS and workflow management into a comprehensive software platform improved COVID 19 surveillance capabilities in Cameroon. The PlaCARD platform demonstrated that it could be used as a LIMS for managing and storing test data during an outbreak.

## Introduction

Global health security is a growing challenge for the 21st century as new pathogens regularly emerge and older ones resurface. Following the emergence of a novel coronavirus (SARS-CoV-2) in Wuhan, China, in December 2019 and its global spread, resulting in unprecedented social and economic costs and loss of many lives [1,2]. Laboratory tests for rapid detection of SARS-CoV-2 by polymerase chain reaction (PCR) have been introduced to ensure appropriate diagnosis and treatment of patients [3]. The first case was confirmed in Cameroon on 05 March 2020, and contact tracing around this first confirmed case led to confirmation of new cases. Within 2 weeks of the identification of the first case, nearly 29 new cases were confirmed. The number of COVID-19 tests increased rapidly, so that one month after confirmation of the first case, approximately 2400 samples were tested with an average of 80 analyzes per day. During the initial phase of the COVID 19 outbreak, the Center Pasteur du Cameroun (CPC) was the only COVID 19 testing center in Cameroon. The increasing demand for COVID 19 laboratory testing around the clock exceeded capacity and placed a significant burden on laboratory staff and infrastructure. To cope with the increase in COVID 19 testing, a strategy to decentralize COVID 19 diagnostics was implemented. This strategy enabled the qualification of 17 additional molecular biology laboratories in nine of Cameroon's 10 regions [4].

In this decentralized environment, it has become essential to harness the power of laboratory information management systems (LIMS) to streamline all phases of laboratory testing (preanalytical, analytical, and postanalytical). Indeed, the COVID 19 pandemic encouraged the use of numerous digital approaches to disease surveillance. It was clear to stakeholders that existing surveillance and response systems based on paper forms, phone calls, text messages, and Excel spreadsheets were far from adequate [5]. Most surveillance systems aimed to expedite the transmission of epidemiologic data over the Internet to mobile devices such as cell phones or tablets. However, none of these approaches included all components and actors involved in disease surveillance and outbreak control strategy in the country. Nor were they able to handle the rapid and multidirectional flow of real time information that is critical to the success of the strategic response plan. The pnademic COVID 19 pandemic highlighted the urgent need to transform the public health system from a reactive to a proactive system and to

develop innovations that provide real-time information for proactive decision making at the local, state, and national levels [6–8].

The rapid increase in the number of COVID 19 tests performed by laboratories resulted in a long turnaround time for laboratory results, leading to delays in confirming the status of suspected cases and initiating immediate action by the case investigation team, delayed communication among response teams, and thus delays in taking immediate action and strategic decisions.

Moreover, one of the first measures the country takes to combat the pandemic is to close the borders [9,10]. After the transmission dynamics were brought under control, some countries reopened their borders. However, international travelers with a negative COVID 19 PCR test could cross the border within 72 hours before departure [11]. This restriction led to falsification of COVID 19 PCR test results by international travelers in several countries, particularly Cameroon. Therefore, COVID 19 testing laboratories faced particular challenges that required specific LIMS capabilities to ensure safe, reliable, and certifiable test results with acceptable turnaround times. However, there is very little literature on tools that use LIMS to reduce laboratory staff burden, streamline laboratory testing, improve test results and certification, facilitate epidemiologic and translational research, and enable data-driven policy decisions.

Based on CPC's experience with biosurveillance, the platform for collecting, analyzing, and reporting data (PlaCARD) [12] was rapidly adapted and linked to the decentralization strategy of COVID 19 testing in Cameroon to support COVID 19 diagnostic laboratories. The main customization concerns the data flow to accommodate the streamlining of lab testing, automating the presentation of test results, integrating QrCode authentication into the test result certificate, and developing an easy-to-use standalone dashboard to visualize the collected data.

In this paper, we describe the architecture and features of PlaCARD and report on the experience of commissioning this platform for the decentralization of COVID 19 diagnostics in Cameroon.

## Methodology

### Data flow

The COVID 19 diagnosis in Cameroon has been progressively decentralized to increase testing capacity, which has increased from less than 100 samples analyzed in the first two weeks of the epidemic to more than 2,000 samples analyzed daily. This led to the establishment of 18 laboratories for molecular diagnosis of COVID in nine of Cameroon's 10 regions and the opening of sampling sites in all health districts of the country. Data were thus collected by the investigation and rapid intervention teams at the sampling sites. The collected samples were sent to the laboratory along with the corresponding data collection sheets. At the laboratory, the sampling sheets were entered into PlaCARD. After the samples were analyzed, the results were entered into PlaCARD by an assistant biologist. The results were then validated and reported by the senior scientist responsible for the laboratory. After validation, users received notification of the availability of their test result, along with a link to download their test result certificate in portable document format (pdf) and the individual download code via SMS. The test result certificate generated by PlaCARD contains an encrypted quick response code (QR code) that can be used to verify the authenticity of the COVID 19 results. This requires reading and decrypting the QR code. A mobile application has been developed to read this encrypted QR Code to verify the authenticity of the COVID 19 test at the border health post level. The results certificate contains the following information's: the first and last name, age and gender of the person tested, the place of sampling, the type of test performed, the date of sampling and the test result.

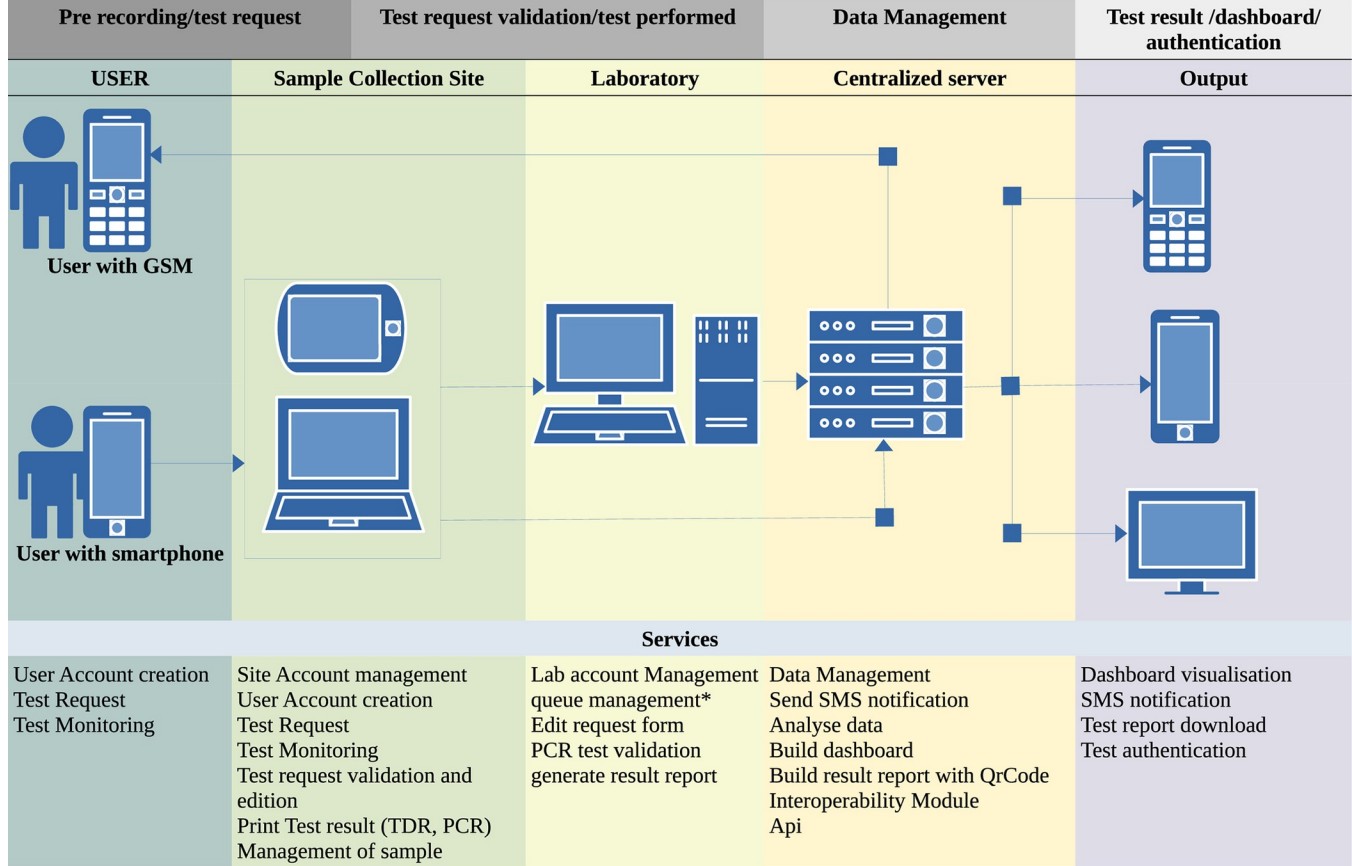

| Pre recording/test request | Test request validation/test performed | | Data Management | Test result /dashboard/ authentication |
|---|---|---|---|---|
| **USER** | **Sample Collection Site** | **Laboratory** | **Centralized server** | **Output** |

**Fig 1. PlaCARD data flow for COVID 19 Diagnostic data management and authentication.**

All COVID 19 diagnostic data were sent to a central database in the CPC that is directly linked to the user-accessible web-based application ([Fig 1]). Based on current and past data stored on the platform, algorithms generate epidemiologic indicators for surveillance when the number of disease cases in a given location or group of people increases beyond expected levels in a given time period. Real-time data and case management capabilities are then used to activate outbreak response actions to bring the outbreak under control. Multidirectional information flows allow the various actors in the national surveillance and response system to receive information and publish new information that is then immediately accessible to all who need to see it.

PlaCARD is designed to improve the efficiency and timeliness of disease control efforts. PlaCARD differs from other digital applications in this area in that it serves as a business process management tool: the entry of a suspected or confirmed case by a data clerk at any level of the system automatically triggers a series of actions to ensure that the case is handled quickly and efficiently.

## Architecture

PlaCARD is an open-source, real-time digital health platform that includes web and mobile applications designed to improve the efficiency and timeliness of disease control interventions. The PlaCARD-based tool facilitates surveillance workflows and automated analysis of key components of routine and active surveillance using protocols recommended by WHO, with

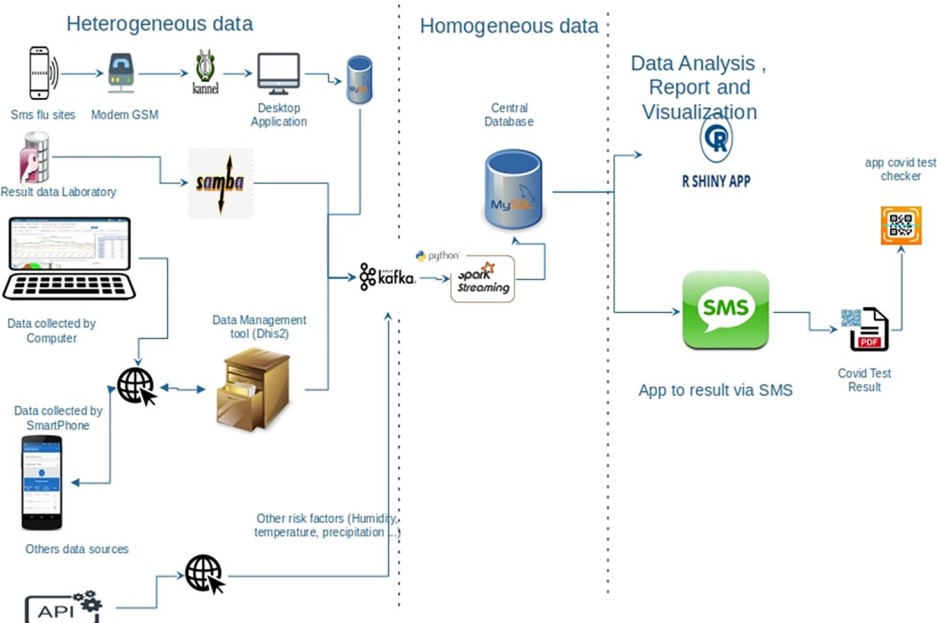

**Fig 2. Architecture of PlaCARD for COVID 19 diagnostic data management and test authentication.**

multiple integrated data sources and applications feeding data into the main surveillance and tracking system. See (Fig 2) for COVID 19 response.

The data come from different sources in different formats. Therefore, they need to be restructured, harmonized, and stored in a central database. In addition, the data from different sources must be taken into account in real time. To achieve this, Apache Kafka and Apache Spark were used to retrieve, process, and store the incoming data streams in a format suitable for analysis. For each type of data collected, a Kafka topic was created and linked to the corresponding data source to make it a producer. The script running on Apache Spark is responsible for consuming the data provides by the producer, transforming it according to the defined processes and storing it in the central database. The purpose of this architecture is to take into account the data collected in the analysis phase in real time.

The PlaCARD platform has been designed to facilitate interoperability with the main digital health applications and national surveillance tools. Application programming interfaces (APIs) have been developed for data exchange between PlaCARD, District Health Information System 2 (DHIS2), and other digital health applications used in Cameroon. You can even set up scheduled integration jobs to synchronize or import data from other sources on a regular basis. The built-in SMS engine enables two-way communication between data collection on mobile devices and the server to receive or send messages.

PlaCARD enables the management of aggregated routine data through a flexible metadata model. Everything can be configured through the user interface: You can set up data elements, data entry forms, validation rules, indicators and reports to create a fully functional system for data management.

PlaCARD uses the R/Shiny Web Framework [13] to develop an easy-to-use, stand-alone dashboard. This application allows selected staff and researchers to quickly query and visualize data without additional informatics support. PlaCARD supports all common chart types such as column, line, pie, stacked column and area charts. PlaCARD has a fully web-based pivot table that allows you to analyze data along all data dimensions and arrange them as needed in

columns, rows and as filters. One can edit data, remove blank rows, and control display density and font size. Pivot tables can be saved as favorites and downloaded.

PlaCARD meets the highest data protection requirements and access to data is through an individually secured account. It includes various user roles within public health services, such as local informants, epidemiologists, laboratory technicians, assistant biologists, senior scientists, and data clerks.

Specially designed dashboards present information tailored to the needs of different stakeholders within the health system. For example, managers at the central level can see at a glance how public health is doing across the country, and dashboards in the emergency operations center play an important role in decision making.

## Implementation

PlaCARD was the available platform for managing COVID 19 diagnostic data in Cameroon. It was linked to the COVID 19 strategy for decentralization of diagnostics in Cameroon. In this way, it was used in all laboratories performing molecular diagnosis of COVID 19 in Cameroon.

For PlaCARD to be used in a laboratory, the laboratory needed a computer to enter the information from the sampling sheet and display the results, an Internet connection to communicate with the central server at the CPC, and a data entry clerk to complete the sampling sheet. Thanks to the support of the development partners, each laboratory was equipped with computer equipment (computer, modem and printer) and data entry personnel according to its analytical capacity, with one data entry officer for at least 100 samples analyzed per day, one computer and one modem per data entry officer.

A total of 18 data entry clerks, 17 computers and printers, and 20 modems were used, and each laboratory had an Internet package to ensure communication with the central server at the CPC.

The implementation phase was accompanied by two days of training of data entry clerks and laboratory personnel in the use of PlaCARD and four hours of training of personnel at the level of each region in the use of the PlaCARD dashboard.

Two full-time data managers were hired to manage the database, interact with the various laboratories to handle requests, and assist them in managing their data.

## Statistical analyses

The outcomes considered in this study were the proportion of tests reported in PlaCARD compared with tests performed in the laboratories, workload, and turnaround time in the laboratories during the study period. Workload was measured by the daily number of tests performed by all laboratories, and turnaround time was the time between the date of sample receipt and the date of the result.

Quantitative results were summarised using the median and interquartile range (IQR) and frequency of qualitative results. The dynamics of the epidemic COVID 19 were graphically represented using the number of PCR tests performed in laboratories and reported in PlaCARD, as well as the positivity rate during the study period. All statistical analyses and graphical representations were performed using R 4.1 software [14].

## Results

PlaCARD was deployed in the CPC to manage COVID 19 data from March 2020 and was then deployed in other COVID 19 molecular diagnostic laboratories as these laboratories were established. From 05 March 2020 to 31 October 2021(all laboratories where Placard has been

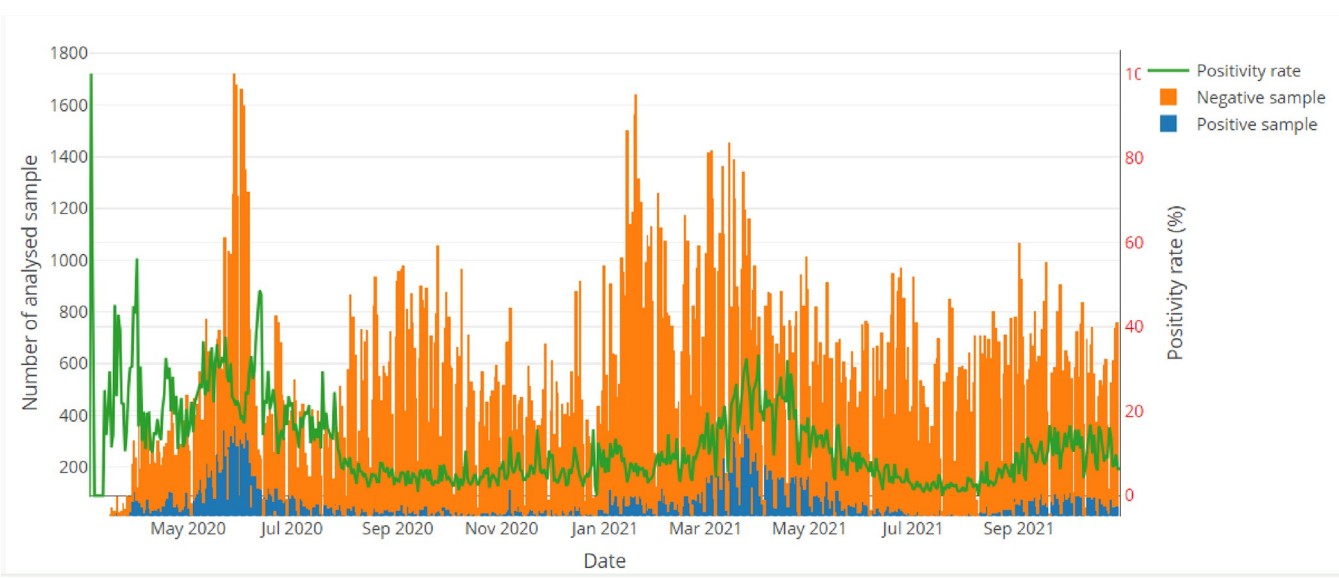

**Fig 3. Evolution of daily number of samples analyzed, positive and positivity rate for COVID 19 test by RT-PCR in Cameroon.**

introduced have used PlaCARD to enter COVID 19 diagnostic data) about 332 000 samples were analyzed and entered into the PlaCARD platform, representing 71% of the number of samples analyzed by molecular diagnostics for COVID 19 in Cameroon.

On average, about 535 (min:132, max:1689) samples were analyzed per day by all laboratories that entered data into PlaCARD. However, testing capacities change over time. An increase in testing effort was observed between May 2020-June 2020 and December 2020-January 2021, corresponding to the high transmission periods of the first and second waves in Cameroon, respectively.

The median turnaround time for results was 1 day [Interquartile Range (IQR): 1–1]. This median turnaround for results delivery was 2 days [0–2.3] before April 2021. It was decreased after the introduction of the SMS results notification system to 1 day [1– 1]. Based on the positivity rate trend, we were able to determine that we have already experienced two waves of the epidemic and are currently in the midst of the third wave. Examination of the curve evolution of the positivity rate showed that the second wave was less important than the first wave in terms of disease transmission (Fig 3). The PlaCARD Dashboard enabled real-time information to be provided to public health decision makers on the spatiotemporal dynamics of the epidemic (Fig 4).

## Discussion

We have shown in this study that PlaCARD ensures a modular and flexible architecture that can be easily adapted and quickly implemented as an emergency LIMS. It is open source and its technical features are constantly evolving to meet changing contexts and user requirements in a context of limited resources.

Experience with integrated disease surveillance and response (IDSR) strategies has shown, paper-based surveillance systems are slow and error-prone [15,16], and these challenges become even greater during epidemics when speed and precision of response are critical. Well-designed mobile and electronic surveillance technologies can overcome many of these drawbacks, feature ease of use and rapid availability of real-time data-essential for managing disease outbreaks in hard-to-reach areas [17,18]. To this end, the PlaCARD platform was

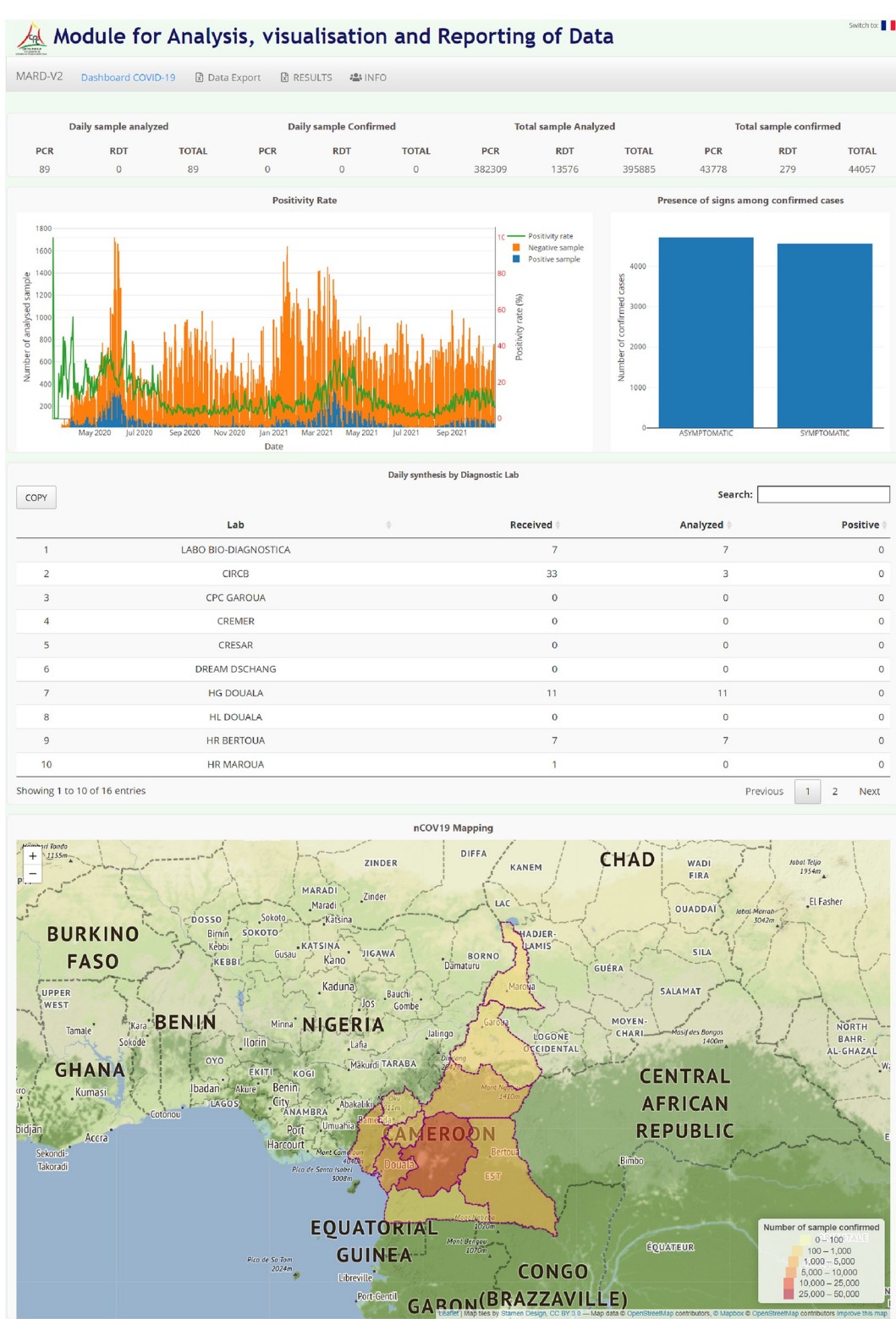

**Fig 4. The PlaCARD real time dashboard at National level.** a) **Basemap source: http://maps.stamen.com/terrain/? request=GetCapabilities&service=WMTS&version=1.0.0#4/19.15/5.27**. b) **Link to the term of use basemap: http://maps. stamen.com/#terrain/4/19.15/5.27**.

developed with the workflow of each laboratory in mind, offering services that reduce the workload of users. In fact, PlaCARD was quickly adopted by laboratories as it enabled automation of results and users could access it from any mobile device or computer with an internet connection through the web-based platform. For this reason, despite the gradual introduction of more than five other digital platforms during the epidemic, PlaCARD remained the most widely used platform by COVID 19 diagnostic laboratories in Cameroon, justified by the 71% share of test data entered into PlaCARD. It is also noted that the proportion of test data entered into PlaCARD is lower than reported in similar studies [19,20] due to the gradual introduction of other platforms during the epidemic. The introduction of the function of sending results by SMS from April 2021 has reduced the turnaround time. Once the test result is validated in the laboratory, the user will receive an SMS with a link to download the test certificate in PDF format. This means that the user no longer has to go to the sampling point or the laboratory to collect their result, and it also reduces queues and the number of people at the sampling point. Several other studies conducted in the context of limited resources [21,22] have shown that sending laboratory results by SMS reduces turnaround time.

Numerous challenges arose during the development, implementation, and support of PlaCARD's end users. Some challenges related to governance, particularly the length of time spent selecting an appropriate LIMS platform given the overwhelming number of solutions available for COVID 19 digital health interventions. Other implementation challenges arose from the poor quality of the data due to the paper-based ordering system. In fact, the paper-based requisition system proved to be a major bottleneck for COVID 19 the sample flow [20]. Important clinical information and contact details were often not provided. Future work will consist of equip PlaCARD with electronic order entry.

Leveraging surveillance and patient data for a real-time evidence-based response COVID 19 response was enabled by PlaCARD. The dashboard allowed selected staff and researchers to quickly query and visualize the data to identify previously unknown correlations, without the need for additional informatics support. However, there is little evidence of actual use of the data for day-to-day decision-making.

We learned several lessons from the PlaCARD system. The first was the difficulty of finding a single platform or solution that would meet all of the COVID 19 monitoring and tracking needs of the country. The DHIS2-based tracker met some, but not all, of the National Emergency Operations Center's LIMS requirements. As a result, the ministry of health and its partners needed to review existing tools in the national eHealth architecture to determine which were best suited for the pandemic and related use cases.

While the solutions identified provided a good foundation for the surveillance and tracking system, they required significant customization to meet local COVID 19 needs and an additional LIMS module. One of the key lessons learned from the Cameroon monitoring system is the importance of local people being able to contextualize and adapt global digital resources to ever-changing local needs. PlaCARD is a local tool developed to fill the gap in some of the LIMS and disease surveillance tools that have been used and can be extended for global disease surveillance. It would be beneficial to maintain such momentum after the crisis COVID 19 to further develop the functions of LIMS that can ultimately increase efficiency, quality and improve employee engagement by eliminating repetitive and administrative tasks.

## Conclusion

The integration of LIMS and workflow management into a single comprehensive software platform (PlaCARD) has strengthened COVID 19 surveillance capabilities in Cameroon by centralizing test data, generating real-time epidemic spread indicators for proactive decision

making, improving turnaround time for COVID 19 test results, and securing test results. The PlaCARD platform has demonstrated that it can be used as a LIMS for managing and securing test data during an outbreak. The addition of the electronic order entry functionality will improve the Platform and will be the subject of future developments

## Acknowledgments

We thank the ministry of health's partners (The WHO country Office, The French Development Agency, the Foreign, Commonwealth and Development Office, US CDC, UNICEF and CHAI) and Cameroon COVID -19 Response Task Force. We also thank the Laboratory Network members who helped implement PlaCARD during the COVID 19 outbreak in Cameroon.

## Author Contributions

**Conceptualization:** Jules Brice Tchatchueng-Mbougua, Achta Hamadou, Serge Alain Sadeuh-Mbah, Paul Alain Tagnouokam-Ngoupo, Richard Njouom, Sara Eyangoh, Mathurin Cyrille Tejiokem.

**Data curation:** Jules Brice Tchatchueng-Mbougua, Loique Landry Messanga Essengue, Francis Jaudel Septoh Yuya, Vanessa Kamtchogom.

**Formal analysis:** Jules Brice Tchatchueng-Mbougua.

**Funding acquisition:** Sara Eyangoh.

**Investigation:** Jules Brice Tchatchueng-Mbougua, Loique Landry Messanga Essengue.

**Methodology:** Jules Brice Tchatchueng-Mbougua, Loique Landry Messanga Essengue, Francis Jaudel Septoh Yuya, Vanessa Kamtchogom, Achta Hamadou, Maurice Tchuente, Mathurin Cyrille Tejiokem.

**Project administration:** Jules Brice Tchatchueng-Mbougua.

**Software:** Jules Brice Tchatchueng-Mbougua, Loique Landry Messanga Essengue, Francis Jaudel Septoh Yuya, Vanessa Kamtchogom.

**Validation:** Jules Brice Tchatchueng-Mbougua, Serge Alain Sadeuh-Mbah, Paul Alain Tagnouokam-Ngoupo.

**Visualization:** Jules Brice Tchatchueng-Mbougua.

**Writing – original draft:** Jules Brice Tchatchueng-Mbougua, Loique Landry Messanga Essengue.

**Writing – review & editing:** Jules Brice Tchatchueng-Mbougua, Loique Landry Messanga Essengue, Achta Hamadou, Serge Alain Sadeuh-Mbah, Paul Alain Tagnouokam-Ngoupo, Maurice Tchuente, Richard Njouom, Sara Eyangoh, Mathurin Cyrille Tejiokem.

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
