## [Decision Letter · Decision Letter 0]

4 Jun 2022

PDIG-D-22-00034

Improving the management and security of COVID 19 diagnostic test data with a digital platform in resource-limited settings: the case of PlaCARD in Cameroon.

PLOS Digital Health

Dear Dr. Tchatchueng Mbougua,

Thank you for submitting your manuscript to PLOS Digital Health. After careful consideration, we feel that it has merit but does not fully meet PLOS Digital Health's publication criteria as it currently stands. Therefore, we invite you to submit a revised version of the manuscript that addresses the points raised during the review process.

Please submit your revised manuscript by . If you will need more time than this to complete your revisions, please reply to this message or contact the journal office at digitalhealth@plos.org. Please include the following items when submitting your revised manuscript:

We look forward to receiving your revised manuscript.

Kind regards,

Jasmit Shah

Guest Editor

PLOS Digital Health

Journal Requirements:

State what role the funders took in the study. If the funders had no role in your study, please state: “The funders had no role in study design, data collection and analysis, decision to publish, or preparation of the manuscript.”

2. In the Funding Information you indicated that no funding was received. Please revise the Funding Information field to reflect funding received.

Please ensure that the funders and grant numbers match between the Financial Disclosure field and the Funding Information tab in your submission form. Note that the funders must be provided in the same order in both places as well.

3. Please update your Competing Interests statement. If you have no competing interests to declare, please state: “The authors have declared that no competing interests exist.”

4. All figures and supporting information files will be published under the Creative Commons Attribution License (creativecommons.org/licenses/by/4.0/). Authors retain ownership of the copyright for their article and are responsible for third-party content used in the article. 

Figure 2: please (a) provide a direct link to the base layer of the map used and ensure this is also included in the figure legend; (b) provide a link to the terms of use / license information for the base layer. We cannot publish proprietary or copyrighted maps (e.g. Google Maps, Mapquest) and the terms of use for your map base layer must be compatible with our CC-BY 4.0 license. 

Please upload any written confirmation as an 'Other' file type. It must clarify that the copyright holder understands and agrees to the terms of the CC BY 4.0 license; general permission forms that do not specify permission to publish under the CC BY 4.0 will not be accepted. Note that uploading an email confirmation is acceptable.

Additional Editor Comments (if provided):

Abstract: Kindly please divide into three sections (Background, Methodology/Principal Findings, and Conclusions/Significance). 

Methodology: Please revise and structure it accordingly. Needs to be more clear. 

Discussion: Please revise the discussion. The Discussion should spell out the major conclusions of the work along with some explanation or speculation on the significance of these conclusions. How can future research build on these observations? What are the key experiments that must be done?

Reviewers' comments:

Reviewer's Responses to Questions

**Comments to the Author**

1. Does this manuscript meet PLOS Digital Health’s publication criteria? Is the manuscript technically sound, and do the data support the conclusions? The manuscript must describe methodologically and ethically rigorous research with conclusions that are appropriately drawn based on the data presented.

Reviewer #1: Partly

Reviewer #2: Yes

2. Has the statistical analysis been performed appropriately and rigorously?

Reviewer #1: No

Reviewer #2: Yes

3. Have the authors made all data underlying the findings in their manuscript fully available (please refer to the Data Availability Statement at the start of the manuscript PDF file)?

Reviewer #1: No

Reviewer #2: No

4. Is the manuscript presented in an intelligible fashion and written in standard English?

Reviewer #1: Yes

Reviewer #2: No

5. Review Comments to the Author

Reviewer #1: The authors of the manuscript “Improving the management and security of COVID 19 diagnostic test data with a digital platform in resource-limited settings: the case of PlaCARD in Cameroon.” Described the relevant topic.

They provided a lot of details about COVID 19 testing in Cameroon.

Improvements are necessary in this manuscript:

1. The entire manuscript is provided as a narrative, without clear separation of headings. Authors should re-write the manuscript and clearly divide Introduction, Methods, Results, etc. 

2. Methodology section is more written as an introduction than standard methodology. The authors should provide methods that they used in the study as well as tests they used to determine the results of the COVID 19 testings. What statistics did they use to determine how significant are findings?

3. Results should provide at least some tables so readers can better understand the text provided in the results section. 

4. Discussion should analyze obtained results and compare them with similar studies. That is lacking in this section.

5. Overall this is an interesting manuscript with the potential for improvements of the text, which is required before publishing.

Reviewer #2: The authors described the architecture, implementation, and requirements of PlaCARD, a software platform to manage patient registration, medical specimens, diagnostic data flow, diagnosis results reporting, and authentication during the 2019 coronavirus pandemic (COVID -19) in Cameroon. The research work reported is of significant practice in the community. 

Some suggestions are listed below to improve the manuscript’s quality. 

1. The representation in the introduction should be further improved, especially by highlighting the authors' contributions. The authors may consider some healthcare-based literature and cite them, e.g., 10.1016/j.knosys.2021.107338 and 10.1109/JIOT.2019.2949715. 

2. The figures are a bit blurry. Please replace them with high-resolution pictures. 

3. The authors may consider analyzing the in-depth reasons hidden in the experimental results.

4. In the conclusion, the future work should be discussed.

5. There are some typos and grammar errors in the manuscript. Please polish the manuscript.

6. PLOS authors have the option to publish the peer review history of their article (what does this mean?). If published, this will include your full peer review and any attached files.

**Do you want your identity to be public for this peer review?** For information about this choice, including consent withdrawal, please see our Privacy Policy.

Reviewer #1: No

Reviewer #2: No

---

## [Decision Letter · Decision Letter 1]

26 Aug 2022

Improving the management and security of COVID 19 diagnostic test data with a digital platform in resource-limited settings: the case of PlaCARD in Cameroon.

PDIG-D-22-00034R1

Dear Dr Tchatchueng Mbougua,

We are pleased to inform you that your manuscript 'Improving the management and security of COVID 19 diagnostic test data with a digital platform in resource-limited settings: the case of PlaCARD in Cameroon.' has been provisionally accepted for publication in PLOS Digital Health.

Best regards,

Jasmit Shah

Guest Editor

PLOS Digital Health

Reviewer Comments (if any, and for reference):

Reviewer's Responses to Questions

**Comments to the Author**

1. If the authors have adequately addressed your comments raised in a previous round of review and you feel that this manuscript is now acceptable for publication, you may indicate that here to bypass the “Comments to the Author” section, enter your conflict of interest statement in the “Confidential to Editor” section, and submit your "Accept" recommendation.

Reviewer #1: (No Response)

Reviewer #2: All comments have been addressed

2. Does this manuscript meet PLOS Digital Health’s publication criteria? Is the manuscript technically sound, and do the data support the conclusions? The manuscript must describe methodologically and ethically rigorous research with conclusions that are appropriately drawn based on the data presented.

Reviewer #1: Partly

Reviewer #2: Yes

3. Has the statistical analysis been performed appropriately and rigorously?

Reviewer #1: No

Reviewer #2: Yes

4. Have the authors made all data underlying the findings in their manuscript fully available (please refer to the Data Availability Statement at the start of the manuscript PDF file)?

Reviewer #1: Yes

Reviewer #2: No

5. Is the manuscript presented in an intelligible fashion and written in standard English?

Reviewer #1: Yes

Reviewer #2: Yes

6. Review Comments to the Author

Reviewer #1: The authors of the manuscript “Improving the management and security of COVID 19 diagnostic test data with a digital platform in resource-limited settings: the case of PlaCARD in Cameroon.” described the relevant topic. They provided a lot of details about COVID 19 testing in Cameroon.

The authors addressed some of my concerns from the initial version but additional improvements are necessary:

Introduction:

1. English needs significant improvement, there are many grammatical errors. For example, this sentence is not clear: “The first case was confirmed in Cameroon on 05 March 2020, and tracing 73 of contacts in the setting of this first confirmed case led to confirmation of new cases.” To my knowledge the first case with COVID19 was not diagnosed in Cameroon. If the authors talk only about the cases in Cameroon, they should write: The first case in Cameroon was confirmed…

2. The sentence between lines 100-104 should be better written and divided into 2-3 shorter sentences, as well as the sentence between lines 113-116 and 117-121. It’s hard to understand them in the current format.

Methodology

1. The authors significantly improved methodology from the initial version of the manuscript. There are similar issues as in the Introduction. English needs improvement. Also, Data Flow section is too long and contains a lot of unnecessary details.

Results, Discussion, Conclusion

1. The title of the manuscript mentions the security of COVID 19 data, but authors didn’t provide any information about security of the data in these 3 sections. The word security should be removed from the title, or the authors should provide the analysis of the security of data and preset results.

2. Statistical analyses were mentioned in the Methodology but the results of these analyses need to be presented better. What is the significance of these analyses, what are the exact numbers for each of analyses, this should be clearly described and systematically presented.

3. Figures and tables are not visible so I cannot read them and cannot make any conclusion about them. They have to be resubmitted in much better resolution so they can be visible to readers.

4. Discussion should compare results with studies from neighboring countries (if they have similar solutions) or with different areas in the world, and show potential advantages of the developed solutions in Cameroon. This comment was not addressed in the previous response.

5. Overall, the current version of the manuscript is significantly improved from the initial version, but it requires additional corrections before publishing.

Reviewer #2: The authors had addressed my concerns.

7. PLOS authors have the option to publish the peer review history of their article (what does this mean?). If published, this will include your full peer review and any attached files.

**Do you want your identity to be public for this peer review?** For information about this choice, including consent withdrawal, please see our Privacy Policy.

Reviewer #1: No

Reviewer #2: No
